# Delivery Strategies for Colchicine as a Critical Dose Drug: Reducing Toxicity and Enhancing Efficacy

**DOI:** 10.3390/pharmaceutics16020222

**Published:** 2024-02-03

**Authors:** Yaran Lei, Yulu Yang, Guobao Yang, Ao Li, Yang Yang, Yuli Wang, Chunsheng Gao

**Affiliations:** 1State Key Laboratory of Toxicology and Medical Countermeasures, Beijing Institute of Pharmacology and Toxicology, Beijing 100850, China; leiyaran375@163.com (Y.L.); yl18176358195@163.com (Y.Y.); yangguobao0101@163.com (G.Y.); liao1746@126.com (A.L.); amms2013@126.com (Y.Y.); 2School of Pharmaceutical Engineering, Shenyang Pharmaceutical University, Benxi 117004, China; 3School of Pharmacy, Guangxi Medical University, Nanning 530021, China; 4School of Pharmacy, Henan University, Kaifeng 475004, China

**Keywords:** colchicine, toxicity reduction and potency enhancement, delivery strategy, transdermal drug delivery

## Abstract

Colchicine (COL), a widely used natural drug, has potent anti-inflammatory effects; however, as a narrow therapeutic index drug, its clinical application is limited by its serious gastrointestinal adverse effects, and only oral formulations are currently marketed worldwide. Recent studies have shown that transdermal, injection, and oral drug delivery are the three main delivery strategies for COL. This article elaborates on the research progress of different delivery strategies in terms of toxicity reduction and efficacy enhancement, depicting that the transdermal drug delivery route can avoid the first-pass effect and the traumatic pain associated with the oral and injection routes, respectively. Therefore, such a dosage form holds a significant promise that requires the development of further research to investigate effective COL delivery formulations. In addition, the permeation-promoting technologies utilized for transdermal drug delivery systems are briefly discussed. This article is expected to provide scientific ideas and theoretical guidance for future research and the exploration of COL delivery strategies.

## 1. Introduction

Colchicine (COL), a tricyclic alkaloid extracted from *Colchicum autumnalis* L. (autumn crocus, meadow saffron) and *Gloriosa superba* L. (glory lily), has been used for thousands of years to treat various diseases and is one of the oldest drugs still available [1]. In 2009, the FDA approved COL for the treatment of gout and familial Mediterranean fever. In recent years, COL has also been shown to have good therapeutic potential in cardiovascular diseases such as pericarditis, coronary artery disease, and acute myocardial infarction [2,3,4]. In June 2023, the US FDA approved 0.5 mg colchicine tablets, LODOCO^®^ (manufactured in AGEPHA Pharma, Vienna, Austria), as the first anti-inflammatory drug for cardiovascular disease, which reduces a confirmed diagnosis of atherosclerotic cardiovascular disease (ASCVD) in adult patients with a cardiac event risk by an additional 31% as compared to placebo [5]. In addition, recent studies have shown that COL has also been used in the treatment of COVID-19, and that its use improves clinical outcomes and reduces hospitalization time in patients with COVID-19 lung infections compared to those receiving placebo or conventional therapy [6].

The therapeutic efficacy of COL Is mainly attributed to its unique anti-inflammatory effects [7]. Unlike nonsteroidal anti-inflammatory drugs (NSAIDs) and steroids that act on the arachidonic acid pathway, it binds to the cytoskeletal protein microtubule protein dimer. Microtubules are involved in various cellular processes, including maintenance of cell shape, intracellular trafficking, cytokine and chemokine secretion, cell migration, and regulation of ion channels and cell division. At low concentrations, COL inhibits microtubule aggregation and affects several cellular processes and pathways modulating the inflammatory response; at high concentrations, it promotes microtubule depolymerization and causes severe toxicity to normal tissues, which limits its use in cancer therapy [8]. COL is involved in multiple mechanisms of inflammatory response regulation based on its disruption of microtubule proteins (Figure 1). In vitro studies have shown that at nanomolar concentrations (50% inhibitory concentration, IC_50_ of 3 nM), COL can reduce the expression of E-selectin on the surface of endothelial cells and inhibit neutrophil adhesion to inflammatory tissues, whereas at micromolar concentrations (IC_50_ = 300 nM), COL inhibits the expression of the adhesion molecule L-selectin on neutrophils, further preventing their recruitment [8,9]. In addition, COL inhibits neutrophil chemotaxis and superoxide production and reduces oxidative stress by decreasing calcium (Ca^2+^) influx into neutrophils [10]. It has been suggested that these biological effects may explain the preventive effect of COL on gout at low doses and the therapeutic effect of COL on acute gout attacks at high doses [11]. Moreover, the inhibition of neutrophil function by COL may be an important reason for its therapeutic effects in cardiovascular diseases [9].

Although COL is a natural product with multiple biological activities and COL formulations have been marketed internationally as early as the 1960s, its toxicity cannot be ignored [12]. COL has been marketed globally as solid oral formulations such as tablets and capsules and oral liquid formulations. However, as a critical-dose drug, the clinical application of COL is limited by its narrow therapeutic window [13]. According to research, severe gastrointestinal adverse reactions, such as bloating, nausea, and diarrhea, occur in nearly 91% of patients [14], and long-term colchicine administration can result in toxicity or organ dysfunction [15]. A recent review article [16] states that the most significant causes of colchicine poisoning are unauthorized access, intentional overdose, and in-appropriate dosing for gout flares. There are currently no specific or effective treatments for colchicine poisoning, and supportive care is relied upon. Safe use awareness for colchicine should be focused on. Therefore, there is a need to design and develop new COL delivery strategies to alleviate adverse reactions and increase its efficacy. In this study, a recent search of COL formulation research revealed that transdermal, injectable, and oral delivery are the three common modes of COL delivery (Figure 2). Among them, transdermal administration is the most widely studied because it effectively avoids first-pass effects, allows stabilization of blood concentrations, reduces gastrointestinal side effects, and drug delivery can be rapidly terminated in the case of serious adverse reactions, making it a very promising route of drug delivery [17]. In addition, the combination of targeted drug delivery systems to effectively deliver COL to the disease site by injection and the preparation of extended-release formulations for oral administration to achieve stable blood concentrations are also effective means of reducing drug toxicity and enhancing efficacy [18]. Therefore, this review classifies and summarizes the above three delivery strategies to provide new ideas and discussions for the future development of novel COL formulations.

## 2. Transdermal Drug Delivery

Transdermal drug delivery often faces challenges in terms of limitations but remains an area with good growth potential. The skin, the largest organ of the body, provides sufficient conditions for transdermal drug delivery; however, the dense “brick-wall structure” of the cuticle makes it difficult for most drugs to penetrate in sufficient quantities to achieve efficacy [17,19]. Even for low clinically effective therapeutic doses of COL, although several studies have confirmed the feasibility of transdermal absorption, the cuticle barrier makes the transdermal absorption of COL inefficient and makes it difficult to achieve the clinical therapeutic effects. Therefore, some means of permeation promotion are still required to facilitate the absorption of the drug into the skin. Therefore, several permeation-promoting techniques have been developed to achieve effective drug penetration, including pharmacological permeation using nanocarriers, chemical permeation using transdermal absorption enhancers, and physical permeation techniques using microneedles [20,21,22]. Among these, pharmacological permeation promotion methods have been studied relatively widely and intensively owing to their high flexibility and targeting. This section summarizes the transdermal delivery of COL using different permeation promotion techniques (Table 1) (Figure 3).

### 2.1. Pharmacological Permeation Promotion

Pharmacological permeation-promoting technologies are mainly based on different permeation-promoting mechanisms, using micro- or nano-drug carriers to improve the permeability of transdermal drug delivery systems by altering the structure or enhancing the mobility of the phospholipid bilayer of the stratum corneum. The pharmacological means of permeation promotion applied to COL include liposomes, lipid nanoparticles, niosomes, lipid liquid crystals, polymeric nanocarriers, and inorganic nanocarriers [36,37,38].

#### 2.1.1. Liposome Technology

Liposomes are tiny closed vesicles formed spontaneously by phospholipids and cholesterol in a dispersion medium and are used as novel drug carriers with properties such as histocompatibility, targeting, and slow-controlled release [39,40,41,42]. Currently, liposome formulations for drug delivery through different routes have been entered into clinical studies or are marketed. Econazole liposome gel was not only the first liposome formulation but also the first transdermal formulation using liposome technology, which was marketed in Switzerland in 1998 [43]. Liposomes can overcome the stratum corneum barrier and facilitate transdermal drug absorption owing to their favorable skin affinity and the advantage of significantly enhancing stratum corneum hydration [44,45,46].

However, it is difficult for traditional liposomes to penetrate the skin stratum corneum because of the low mobility of the lipid layer, strong membrane rigidity, limited permeation promotion, and formation of lipid films on the skin surface after dehydration, resulting in reduced transdermal penetration of drugs [20,47]. To overcome these issues, traditional liposomes have been structurally and surface-modified to increase their elastic deformability, permeability, and stability according to the purpose of drug administration [48,49,50]. According to the literature, the liposomes used for the transdermal absorption of COL include elastomeric liposomes, alcoholic liposomes, and alcohol delivery bodies, whose deformability makes them more favorable for dermal drug administration.

##### Elastic Liposomes

To improve the deformability of conventional liposomes and, therefore, enhance their transdermal penetration, researchers have added edge activators [51,52,53]. Hardevinder et al. [23] prepared a COL elastomeric liposome (COL-ELP) formulation with a particle size of 135 ± 11 nm and an encapsulation rate of 66.3 ± 2.2% using soy phosphatidylcholine (PC) as the membrane material and Span 80 as the edge activator using a conventional rotary evaporation sonication method. Compared with COL conventional liposomes (encapsulation rate of 33.4 ± 2.8%) and COL lipid-like vesicles (encapsulation rate of 31.6 ± 1.4%), COL-ELP exhibited a higher encapsulation rate due to higher vesicle membrane elasticity (*p* > 0.05), and higher lipid composition. The results of in vitro percutaneous permeation experiments in rat abdominal skin showed significant differences in the permeation flux of COL after 24 h of application of different preparations, with COL-ELP having a permeation flux of 44.4 ± 1.9 μg/cm^2^·h, which was approximately 1.5, 1.7, and 10.2 times higher than that obtained with COL conventional liposomes, lipid-like vesicles, and drug solutions. In addition, COL-ELP has better reservoir formation and penetration in deeper skin layers, with an 8-fold increase in skin retention and a nearly 13-fold increase in skin penetration depth (up to 200 μm) compared to drug solutions. Evaluation of the anti-gout activity of different COL preparations in the monosodium urate (MSU)-induced rat air sac model revealed that COL-ELP showed better anti-gout activity in terms of exudate volume, white blood cell count, pathological section condition, and percentage reduction in collagen deposition.

Although the advantages of elastomeric liposomes as transdermal delivery vehicles for COL have been demonstrated in facilitating skin penetration, researchers have found that COL-ELPs have a low drug encapsulation rate and are prone to drug leakage during storage. To solve this problem, Hardevinder et al. [23] prepared COL-cyclodextrin inclusion complexes (COL-βCD) using the freeze-drying method and encapsulated them in a pre-optimized elastomeric liposome prescription (PC:Span 80 = 85:15) to prepare a COL-βCD elastomeric liposome formulation with a drug content of 0.2% *w*/*w* (COL- βCD-ELP). The encapsulation rate of COL in this system increased to 79.5 ± 1.6%, probably due to the hydrophobicity of β-cyclodextrins, which resulted in better retention of COL-βCD inside the elastic liposomes, but this also led to an increase in particle size (152 ± 14 nm) and a significant decrease in permeation flux (26.4 ± 1.2 μg/cm^2^·h). However, the authors noted that for the effective treatment of gout, as well as for other dermatological diseases such as psoriasis and leukocytoclastic vasculitis, whose dosage should be based on skin accumulation of the drug, the skin deposition of COL-βCD-ELP increased 12.4-fold compared to the drug solution after 24 h of administration, which may be due to the additional barrier formed by the release of the drug from the vesicle membrane due to complexation. In addition, COL-βCD-ELP showed better and sustained anti-gout activity in vivo.

##### Alcohol Plasmid

An alcohol plasmid is a liposome with a high ethanol content composed of phospholipids, 20–50% ethanol, and water. It combines the advantages of ethanol and liposomes with strong deformability, a small particle size, a high encapsulation rate, and a high drug-loading capacity. These characteristics enable alcohol plasmids to effectively carry drugs through the stratum corneum to achieve therapeutic effects [54,55,56]. In particular, ethanol acts as a skin penetration enhancer, disrupting the lipid structural domains of the skin stratum corneum, lowering the barrier of the stratum corneum, and allowing alcoholic plasmids to cross the stratum corneum, accumulate in the deeper layers of the skin, and be taken up by skin cells [57,58].

Zhang et al. [24] prepared ice-chip-modified COL alcohol plasmids by grafting ice chips onto dioleoyl phosphate ethylenediamine using ice chips as permeation enhancers. Alcohol plasmids with single-loaded COL and ice chips co-loaded with COL were used as controls. The in vitro transdermal rates of COL of the ice chip-modified alcohol plasmid, co-loaded alcohol plasmid, and single-loaded alcohol plasmid were 5.13, 4.08, and 2.86 times higher than those of the drug ethanol solution, respectively. After transdermal administration of 0.572 mg/kg to rats, the in vivo plasma area under curve (AUC) of the above alcohol plasmid groups were 18.74 ± 2.56 ng/mL, 14.56 ± 2.08 ng/mL, and 14.22 ± 4.39 ng/mL, respectively; 11.37 ± 0.14 ng/mL and 11.63 ± 2.33 ng/mL for the ethanol solution and oral formulation groups, containing the alcoholic plasmid group of ice chips, was significantly higher than the ethanolic solution group (*p* < 0.01). In addition, compared with the control group, the ice chips-modified alcoholic plasmid group showed better inhibitory effects on the inflammatory model of rat toe swelling and stronger down-regulation of the inflammatory factors prostaglandin E2 (PGE-2) and tumor necrosis factor-α (TNF-α), among which the efficacy of ice chips-modified alcoholic plasmid was also better.

##### Alcohol Transferosomes

Transferosomes were the first reported modified liposomes, which improved their transdermal permeation by adding nonionic single-chain surfactants, such as bile acid salts, to conventional liposomes as lipid membrane softeners to increase membrane fluidity and elastic deformation capacity [59]. It can cross the cellular interstices of the stratum corneum to enter the deeper tissues of the skin. When it is administered in an open environment, the osmotic pressure generated by the evaporation of water from the skin is the driving force that squeezes it into the skin through its natural fissures. When it is administered in a closed environment, the driving effect of osmotic pressure is weaker due to the wetness of the skin, and the drug remains in the surface layer of the skin, similar to the slow release of the drug from a reservoir [45,57]. Alcohol delivery bodies are a new type of liposome developed based on delivery bodies and alcohol plasmids, which mainly consist of phospholipids, ethanol, water, surfactants, or permeation enhancers (e.g., cetyltrimethylammonium bromide, stearamide, sodium stearate, deoxycholic acid, sodium taurocholate, oleic acid, and Tween) [60]. Compared with delivery bodies, they improve transdermal drug delivery in both confined and unconfined situations. Compared to an alcohol plasmid, it improves the ability of the encapsulated drug to reside in the deeper tissues of the skin. Thus, alcohol delivery bodies have advantages over alcohol plasmids and delivery bodies for transdermal drug delivery [60,61,62].

Ibrahim M et al. [25] prepared three types of COL alcohol transporters: transethosomes with Tween 20^®^ (TET), transethosomes with sodium taurocholate (TENa), and ransethosomes with Labrafil^®^ (TEL) using PL90G (containing > 90% soy phosphatidylcholine), ethanol, and three different surfactants (Tween 20^®^, sodium taurocholate, or Labrafil^®^) as membrane materials using the cold method and statistically optimized them using a three-group 24 full factorial design experiment. Single-factor experiments showed that the particle size of all alcohol transporters decreased with increasing concentration in the range of 10% to 35% ethanol concentration and increased with increasing phospholipid concentration, and the alcohol transporters prepared using the hydrophilic surfactants Tween 20^®^ and sodium taurocholate (TET and TENa) exhibited smaller particle sizes. Afterwards, the four experimentally preferred alcohol transfer agents were loaded with COL at 0.3% *w*/*w* and dispersed in Carbopol 940 gel matrix (2% *w*/*w* Carbopol 940^®^; 1% triethanolamine; 0.02% nipagin methyl ester; and 0.01% nipagin propyl ester), and the final concentration of COL in all gel formulations was 0.24% *w*/*w*. Considering that formulations with higher static yield values were more stable by maintaining vesicles in suspension and minimizing precipitation, the two regimens with higher yield values were selected for in vitro skin permeation studies. The results of the study showed that the permeation flux of COL from the alcohol transfer body gel (TE gel) was significantly higher than that of the non-alcoholic plasmatic gel (NE gel), which was attributed to the presence of ethanol, phospholipids, and surfactants that increased vesicle ductility and skin lipid perturbation. Compared to the TENa gel, due to the higher vesicle deformability provided by Tween 20^®^ than sodium taurocholate, the TET gel showed a higher permeation flux. Moreover, the preferred final formulation (2% PL90G, 0.2% Tween 20, and 35% ethanol) exhibited satisfactory stability and provided an alternative route for COL administration. Although the application of liposome technology in drug delivery is becoming increasingly widespread, the safety of various types of liposomes and their excipients should still attract sufficient attention, and their long-term toxicology and mechanisms of action need to be systematically and thoroughly studied. In addition, the production process, stability, and sterilization protocols are the main issues that need to be resolved for the industrialization of liposomes.

#### 2.1.2. Niosomes Technology

Vesicles are self-assembled unilamellar or bilamellar vesicular nanocarriers formed by the hydration of nonionic surfactants and cholesterol. Although the structure is similar to that of liposomes, the raw materials are cheap and easy to obtain, and they can overcome the problems that exist in liposomes, such as sterilizability and physical stability. In recent years, they have been developed as an alternative delivery system to liposomes, and they not only can accomplish dermal delivery but also can realize multi-functional drug delivery for a wide range of applications, such as brain-targeted delivery, ophthalmology delivery, and topical vaccine delivery [63].

Elsewedy et al. [26] prepared colchicine vesicles with a nonionic surfactant span 60 and cholesterol 1:1; then sodium alginate was used as the hydrogel matrix, Tween 80 as the aqueous phase, and jojoba oil as the oil phase, which was mixed using a mixer to obtain jojoba oil-based emulsion, and finally the colchicine vesicle formulation was mixed in to obtain colchicine vesicle emulsion gel formulation. After optimizing the prescription using response surface methodology, the final colchicine vesicular emulsion gel formulation was prepared with a particle size of 220.7 nm, a PDI value of 0.22, an encapsulation rate of 65.3%, a pH value of 6.73, and an in vitro release of 52.4%. Jojoba oil itself has anti-inflammatory properties, and the latex formulation (emulgel) makes a transdermal drug delivery system consisting of a gel and an emulsion that can enhance skin permeability and thus drug efficacy. Carrageenan-Induced Rat Paw Edema Test showed that compared to the group using oral colchicine and colchicine vesicle gel, the group using colchicine vesicle emulsion formulation treatment group of rats hind paw inflammation was significantly reduced and paw swelling was reduced to 22.8%, confirming that the addition of jojoba oil and colchicine to the vesicle emulsion could achieve local delivery effects, exert synergistic anti-inflammatory effects, and improve drug utilization.

#### 2.1.3. Lipid Liquid Crystal Technology

Lipid liquid crystals are gel-like intermediate phases formed by the emulsification of lipids at high energy [38]. Lipid liquid crystals are classified into two types: thermogenic liquid crystals and solvogenic liquid crystals, depending on whether the phase change into the liquid crystal state is caused by a change in temperature or by the addition of a solvent. Thermogenic liquid crystals generally exist only in a single substance caused by temperature change, and solvogenic liquid crystals are affected by changes in the concentration of the substance and changes in structure because of changes in surfactant concentration and are classified into lamellar, cubic, and hexagonal phases [64,65]. Cubosomes (Cubs) are of great interest in the field of drug delivery because of their unique internal structures and excellent physicochemical properties [66,67,68]. It is a closed lipid bilayer “honeycomb (spongy)” liquid crystal with bicontinuous aqueous and lipid regions [69], which has high drug loading capacity and pro-permeability, good bioadhesive properties, and the ability to load different drugs simultaneously.

Nasr et al. [27] loaded COL into cubosomal nanoparticles formed by glyceryl monooleate (GMO) and water for transdermal drug delivery. The COL cubosomes were firstly screened and optimized by Box–Behnken analysis factor design and then used for in vitro transdermal permeation experiments through rat skin, and the results showed that the COL transdermal amount of the optimized formula could reach 1307 ± 85.17 μg/cm^2^, which was much higher than the COL transdermal amount of the drug solution (739.13 ± 274.37 μg/cm^2^). In addition to the unique structure of cubosomes, the presence of additional agents such as GMOs and P407 may also be responsible for the enhanced skin penetration of the drug, as they can reversibly disrupt the lipid arrangement of the stratum corneum. Consistent with this, the carrier was combined with a transdermal delivery system to administer a single dose of 0.03 mg/kg of COL cube gel and COL oral solution to rats. As a result, the relative bioavailability of the transdermal COL cubosomes was 4.6 times higher than that of the COL oral solution, effectively improving the bioavailability of COL and reducing its side effects.

Although cubic liquid crystals are extremely promising owing to their unique properties, the mechanism of their permeation-promoting effect is mainly inferred based on specific experimental results, which may lead to opposite permeation-promoting effects with the same physicochemical properties and a lack of systematic research. In addition, cubosomes are not the final formulation, and whether the addition of other excipients will affect the liquid crystal structure and drug efficacy when the formulation is formed remains to be studied; therefore, more in-depth and systematic research is needed for its successful clinical application.

#### 2.1.4. Lipid Nanoparticle Technology

Lipid nanoparticles are colloidal dispersion systems prepared from solid lipids or mixtures of solid lipids and liquid lipids as substrates, which are divided into two types, solid lipid nanoparticles and nanostructured lipid carriers, depending on the different substrate materials, and have the advantages of strong lipophilicity, small particle size, and strong skin adhesion, which can significantly promote the transdermal absorption of drugs [70,71,72].

Joshi et al. [28] prepared and optimized solid lipid nanoparticles (COL-SLNs) loaded with COL by ultrasonication using K-glycerol monostearate (KGMS) as the lipid material and Tween 20 as the surfactant. The COL-SLNs prepared according to the optimized prescription had an average particle size of 107 nm, a zeta potential of −17.4 mV, an encapsulation rate of 37.25%, and were stable at room temperature for one month. In vitro transdermal experiments showed that the cumulative amount of drug permeation could reach 330.34 ± 12.36 μg after administration of COL-SLNs for 48 h. Further, COL-SLNs were made into transdermal patches suitable for dermal administration, and in vivo pharmacokinetic studies showed that the AUC_0→∞_ of the COL-SLNs transdermal patch formulation increased by 2.84 times compared to the free COL patch, and the plasma drug concentration persisted for more than 24 h. Pharmacodynamic studies showed that the COL-SLN transdermal patch significantly reduced the exudate volume in the rat air sac model compared to the free COL patch, and its reduction further increased to 19.42 times that of the free COL patch at the end of 24 h. Similar results were observed in the leukocyte count. The anti-gout potential of the COL-SLNs transdermal patch formulation was further confirmed by staining with eosinophilic hematoxylin.

In recent years, materials with high safety and delivery efficiency have been developed for the preparation of lipid nanoparticles, such as high-affinity lipids and natural derivatives of surfactants, and in combination with other transdermal technologies. These will be new directions for the research of lipid nanoparticles in the field of transdermal drug delivery. In addition, the transdermal transport mechanism of lipid nanoparticles and the development of efficient large-scale production processes are worth exploring as major research directions.

#### 2.1.5. Natural Nanocarrier Technology

Exosomes, as natural and novel nano-delivery vehicles, have attracted much attention from the scientific community because of their ability to regulate multicellular biophysiological functions in a cell-to-cell delivery mode and are also known as “natural lipid nanoparticles” [73]. Exosomes are a subgroup of extracellular vesicles approximately 40–150 nm in diameter, encapsulated by a lipid bilayer membrane, and secreted by most eukaryotic cells [74]. Due to its innate stability, low immunogenicity, biocompatibility, and good biofilm permeability, it plays an essential role in drug delivery; the function of exosomes in clinical diagnosis is also significant, which indicates the pathophysiological conditions through various biomolecules of the host cells and thus detects biomarkers that help diagnose diseases [73]. It has been shown that plant vesicles exhibit excellent functions in drug delivery, especially in the treatment of inflammatory diseases [75]. For example, plant exosome-like vesicles and grapefruit-derived nanovesicles [76] can ameliorate intestinal inflammation; ginger-derived nanovesicles [77] have been shown to significantly reduce the release of inflammatory factors in the gingiva, which is expected to be used against chronic periodontitis. The exosomes deliver small-molecule drugs that have enormous potential for use in preventing the occurrence of disease and treating it.

#### 2.1.6. Polymer Nanocarrier Technology

Polymeric nanocarriers are a class of nanocarriers prepared using polymeric materials with excellent biocompatibility, such as dextran, chitosan, cyclodextrins, and synthetic polymers [78,79,80]. They can be further classified into different types, such as polymer nanoparticles, polymer micelles, nanogels, and dendrimers, based on their structural characteristics [81,82]. They have been increasingly applied in the field of drug delivery owing to their advantages in increasing the solubility of insoluble drugs, improving drug permeability, and regulating the release rate of drugs [83]. Notably, the nanosize of the carriers enhances cellular interactions and permeability through a variety of biological barriers, providing favorable conditions for transdermal drug delivery [84].

Poonam Parashar et al. [29] prepared and optimized chitosan nanoparticles containing COL using a spontaneous emulsification method with a 23-dialysis factor design, and the optimized formulation (CHNP-OPT) had an average particle size of 294 ± 3.75 nm, an encapsulation rate of 92.89 ± 1.1%, a drug content of 83.45 ± 2.5%, and a cumulative drug release within 24 h of 89.34 ± 2.90%. CHNP-OPT was dispersed into 4% *v*/*v* of HPMC E4M polymer to produce chitosan nanoparticle gels containing COL. The nanoparticle gel showed a cumulative in vitro penetration of 74.65 ± 1.90% over 24 h and great stability for over 90 days. The nanoparticle gel exhibited significantly lower uric acid levels in the MSU-induced gout model in rabbits than the normal COL gel, which, combined with the X-ray diffraction results, indicated more desirable anti-gout activity.

Farzaneh Sadeghzadeh et al. [30] incorporated a dual emulsion-evaporation solvent method (DLS) to load COL into polyethylene glycolized poly (lactic-co-glycolic acid) (PLGA) nanoparticles (COL-PP-NPs), and then combined chitosan and folic acid (FA) on the surface of nanoparticles (COL-PPCF-NPs). Consequently, chitosan and folic acid (FA) were bound on the surface of the nanoparticles (COL-PPCF-NPs), delivering COL to HT-29 cancer cells in the colon to induce mitotic-mediated apoptosis. DLS results showed that COL-PPCF-NPs had an average particle size of 250.34 nm, a PDI of 0.32, and a surface charge value of 34.07 mV, with a homogeneous particle size fraction and good stability. The in vitro drug release results showed that the COL release from the nanoparticles was slow, without a sudden release phenomenon, releasing only about 50% of COL from COL-PPCF-NPs at 96 h. The toxicity assay on colon cancer cells showed that COL-PPCF-NPs at a concentration of 250 μg/mL were able to inhibit more than 98% of the cancer cells and selectively bind to folate receptor-positive HT-29 cancer cells, reducing the inhibitory effect on normal cells and increasing the safety of clinical application.

However, the preparation of nanocarriers is more complicated than that of traditional transdermal drug delivery formulations, and expensive polymeric materials increase costs. In addition, during processing into clinically appropriate drug formulations, nanocarriers may undergo particle size changes and drug leakage owing to the addition of excipients, which increases the difficulty of their quality control. All these issues can affect clinical efficacy, and with the resolution of these issues, polymeric nanocarrier technology can be more widely used to benefit patients in the near future.

#### 2.1.7. Inorganic Nanocarrier Technology

In recent years, many inorganic nanomaterials with high drug-carrying capacity, high stability, and good biocompatibility have been discovered, such as mesoporous silica nanoparticles (MSNs), gold nanoparticles (GNPs), graphene oxide (GO), and black phosphorus (BP), which are easy to prepare, have good shape and size controllability, and their surface can be easily modified; therefore, they have great potential for designing drug delivery systems [85]. Mesoporous silica nanomaterials are a class of inorganic porous polymeric nanomaterials with particle sizes ranging from 50 to 200 nm, pore sizes between 2 and 50 nm, regular pore structures, high specific surface areas and pore volumes, and excellent biocompatibility. They have gained wide attention and applications in the field of transdermal drug delivery [86,87,88].

Mohamed et al. [31] prepared mesoporous silica nanoparticles (COL-MSNs) containing COL and encapsulated them in a self-healing hydrogel prepared by the reaction of hydroxyethyl chitosan and prululan oxide to form a composite drug delivery system consisting of COL, inorganic nanoparticles, and hydrogel (COL-MSN-hydrogel system). The particle size of COL-MSNs was determined by dynamic light scattering (DLS) to be 167.1 ± 51.36 nm with a potential of 5.83 mV and a COL encapsulation rate of 67.3 ± 3%. In vitro skin permeation experiments on mouse dorsal skin revealed that the COL hydrogel system showed significantly enhanced drug flux compared to the free COL aqueous solution, and the COL-MSNs hydrogel system with nanocarriers showed even greater enhancement than the COL hydrogel. In addition to the nano-size of COL-MSNs, the large pore volume, and the large surface area, the bioadhesive properties of the homemade hydrogels, the biopolymers used, and the occlusive properties of the hydrogel-loaded cotton bandages contributed to the enhanced penetration of the skin membranes due to the increased hydration of the squamous layer of the skin epidermis. The preparation was loaded onto cotton gauze to form an easy-to-use bandage to ensure accurate drug dosing and applied to the ankle joints of the MIA-induced mouse OA model for treatment. The treatment resulted in increased motor activity and glutathione levels and significantly decreased levels of malondialdehyde, nitric oxide, TNF-α, and COX-2 in the experimental animals, revealing that the developed colchicine mesoporous silica nanoparticle/hydrogel patch is a highly effective COL delivery system while showing great potential in providing an efficient and safe formulation for OA management.

However, inorganic nanocarriers applied in the field of transdermal drug delivery still have potential safety issues, and a study by WU et al. showed that small-sized nano-TiO_2_ can cross the skin stratum corneum and accumulate in the body, causing damage to the skin and internal organs [89]. In addition, heavy metals and organic solvents remaining during the preparation of inorganic nanomaterials can also lead to toxicity. Therefore, it is necessary to conduct a systematic and in-depth study on the dermal toxicology of inorganic nanomaterials to provide a theoretical and experimental basis for their application in the field of transdermal drug delivery.

### 2.2. Chemical Permeation Promotion

Techniques promoting chemical permeation include the use of chemical absorption enhancers and ion pairs. Chemical absorption enhancers are the preferred and most used methods to improve the transdermal absorption of drugs [90,91]. Pressure-sensitive adhesive dispersible (PSA) transdermal patches for transdermal drug delivery are the current focus of development encouraged by regulatory agencies in various countries owing to their simple structure, low process cost, and better patient compliance. A COL-drug-in-adhesive dispersible patch (COL-DIA patch) has been prepared in advance by our group for the transdermal delivery of COL to avoid the side effects associated with oral administration [32]. Based on in vitro permeation experiments on porcine skin (Figure 4), the key excipients (such as PSA) and drug loading in the patch were optimized. To solve the problem of low drug absorption, the permeation enhancer (oleic acid [OA] + propylene glycol [PG]) used in the formula was screened and optimized. The reversible perturbation of stratum corneum lipids by both OA and PG, as well as the synergistic effect between the two permeation enhancers, effectively enhanced 2.01 times the skin permeation flux of COL compared to that of the control group (without the addition of a permeation enhancer). The cumulative in vitro penetration of COL in the preferred formula for 48 h could reach 235.14 ± 14.47 μg/cm^2^. Compared with the oral COL tablet solution, the prepared patch could obtain more stable plasma levels and sustained drug release after transdermal administration, as well as favorable efficacy in an acute gouty arthritis model in rats, effectively reducing adverse drug reactions and prolonging drug activity, making it an attractive alternative route of drug delivery. Inevitably, however, a common problem with the application of chemical promoters is skin irritation. In our experiments, the rat skin exposed to the preferred patch did not show any erythema or edema. However, additional and more complete safety evaluation experiments are still needed.

### 2.3. Physical Permeation Promotion

Physical permeation-promotion techniques include microneedle techniques, ion introduction, electroporation, and ultrasonography. The development of physical permeation promotion techniques in recent years has effectively expanded the range of drugs that can be used for percutaneous absorption and has received increasing consideration [92,93,94]. Liu et al. [33] prepared a COL-loaded soluble microneedle array (COL-MN) with hyaluronic acid (HA; 340 kDa) to produce micron-sized cavities in the skin stratum corneum by puncturing the microneedles to introduce COL (Figure 5). In vitro permeation tests showed that COL-MN had a 3.36-fold higher drug penetration rate than COL gels loaded with equal amounts of active ingredients (Figure 6) and that rats administered COL-MN maintained effective blood concentrations for more than 12 h, 50% longer compared to gavaged COL solutions, and exhibited significant anti-inflammatory effects comparable to the oral route within 5 h.

Yao et al. [34] prepared a soluble microneedle capable of rapid separation to simplify long-term gout management. By preloading urate oxidase (UAO) into liposomes (LPO) and then co-loading it with COL into a slow-release microneedle to obtain (UAO-LPO/COL-MN), the use of chitosan, a microneedle formulation material with a slow-release effect, considerably increased the safety of topical administration of colchicine at low doses. The in vitro permeation test and in vivo retention test showed that UAO-LPO/COL-MN displayed an almost constant drug permeation rate with a cumulative permeation percentage of 71.17 ± 2.40% over 7 days; the uricase activity remained above 90% after 2 months of storage at 4 °C, which improved the stability of uricase and exerted its potent anti-inflammatory effects. In the acute gout model in rats, the slow-release microneedles reduced inflammatory cytokines to different degrees (*p* < 0.05) and led to the rapid reduction in joint swelling, effectively preventing gouty attacks.

In addition, the rise and development of hydrogel microneedles (HMNs) is gradually becoming a novel tool for COL delivery. A recent study reported that Suping Jiang et al. [35] prepared COL-loaded HMNs (COL-HMNs) for the therapeutic management of acute gouty arthritis (AGA) using UV-responsive cross-linkers with UV-responsive disulfide bonds and in situ photopolymerization and that the HMN’s super solubilization capacity improved COL utilization and drug loading, and it had good biological compatibility. Ex vivo experiments demonstrated that the in vitro cumulative release rate of COL-HMNs was as high as (81.50 ± 5.97%) within 48 h. The sustained release of COL on rat skin reduced the level of inflammatory cytokines in rats in an acute gout model, and the development of COL-HMNs provides a promising drug delivery system for the therapy of AGA.

Although the utilization of microneedles has largely increased the transdermal absorption of the drug and reduced its systemic side effects, the length and hardness of the needle inserted in the body, as well as the possibility of infection induced by microorganisms entering the tissue from the open channel due to prolonged use, are issues that should be examined during the study of microneedle technology.

## 3. Oral Administration

Oral administration remains the most used mode of drug delivery owing to its accessibility, ease of administration, and low production costs. However, the currently marketed dosage forms of COL are rapid-release oral formulations, which have problems such as large fluctuations in blood concentration and can lead to serious adverse reactions. Compared to rapid-release formulations, sustained-release formulations can effectively avoid the peak and trough phenomenon of blood drug concentrations, thereby reducing the toxic side effects of drugs. Among these, micropills, as a multi-unit sustained-release formulation, can be widely distributed throughout the gastrointestinal tract with a large absorption area, avoiding irritation caused by high local drug concentrations. Three COL-sustained-release micropills (COL-SRPs) with different in vitro release rates were prepared by adjusting the content of the porogenic agent hydroxypropyl cellulose (HPC; 15%, 22%, and 25%) using fluidized bed coating technology [95]. The microscopic morphology, drug loading, in vitro release, and mechanism of action of the micropills were characterized, and the results showed that the release rate of COL-SRPs increased with increasing HPC content. Consistent with this, after a single oral dose of COL-SRPs in rats, sustained-release micropills with 15% HPC exhibited better in vivo sustained-release properties, with significantly higher Tmax, and mean residence time (MRT) values than micropills and COL tablets (COL-Ts) with other HPC contents. Similar to the results of in vivo pharmacodynamic and hepatic and renal function index studies in rats, showed that compared to COL-Ts, COL-SRPs (15%) had better anti-inflammatory effects and could be safely and effectively used for the long-term prevention of gout (Figure 7).

## 4. Injection Administration

### 4.1. Intravenous Injection

After injection, the drug can enter the blood vessels or tissues directly, which can greatly shorten the absorption time with a rapid onset of action and direct effect. However, intravenous injection leads to frequent clinical adverse events due to poor safety and body adaptability. Intravenous injection of COL was banned by the Food and Drug Administration in 2008 owing to serious adverse consequences, including approximately 2% mortality [13]. In recent years, the combination of targeted drug delivery systems using nanocarriers and injectable drug delivery to selectively act on target organs or cells has received wide attention in terms of improving efficacy and reducing toxic side effects. KA et al. [96] used phosphonate-functionalized MSNs functionalized and loaded with COL and subsequently coated them with chitosan glycine coupled with FA to obtain MSNsPCOL/CG-FA nanoformulations for targeting COL in colon cancer cells. Compared to free COL, targeted administration of COL achieved highly enhanced anticancer effects, with up to 100% inhibition of HCT116 colon cancer cells. Meanwhile, the cytotoxicity of MSNsPCOL/CG-FA (4%) was much lower than that of COL (~60%) in BJ1 normal cells, providing a new approach for developing cancer immunotherapy using low-cost small-molecule natural prodrugs.

Li et al. [97] utilized modified macrophage membranes encapsulated with COL-containing PLGA nanoparticles (MMM/COL-NPs) for the local delivery of COL to effectively prevent atherosclerotic plaque formation. In vitro experiments showed that the highly expressed integrin α4/β1 on the modified macrophage membranes can effectively target endothelial cells, thus mediating the recruitment of MMM/COL-NPs to atherosclerotic plaques; meanwhile, the membrane CD47 protein has superior immune escape properties and can allow evade clearance by macrophages. Intravenous injection of this complex system into an atherosclerosis mouse model significantly reduced plaque area and improved plaque stability, while significantly decreasing macrophage infiltration (macrophage immunofluorescence staining of frozen sections in the untreated group accounted for 71.20 ± 5.44% of the whole aorta, compared to 31.39 ± 4.29% after the administration of MMM/COL-NP) (Figure 8). This delivery system provides not only a new solution to achieve effective delivery of COL but also an effective and safe method for the treatment of atherosclerosis and may even be extended for the treatment of other inflammatory diseases.

Vascular inflammation and oxidative stress contribute to cardiovascular disease [98]. Recent studies have shown that inflammation is closely related to stress oxidation. Risk factors that predispose to cardiovascular disease, such as smoking, alcohol abuse, and hypertension, produce oxidative stress, which subsequently leads to DNA damage, resulting in cellular senescence and the senescence-associated secretory phenotype (SASP) [99]. And SASP increases the cellular inflammatory factors, chemokines, and release of adhesion molecules and matrix metalloproteinases, which exacerbate inflammation and cause tissue remodeling, leading to the formation and progression of atherosclerotic plaques. Oxidative stress-induced DNA damage activates the NF-κB, MAPKs, and mTOR pathways, and colchicine has a protective effect against cardiovascular disease due to its ability to block oxidative stress-induced endothelial cellular senescence through inhibition of the NF-κB and MAPKs pathways, inhibit inflammation, and reverse SASP [100].

In addition, a recent large-scale clinical study confirmed that low-dose COL significantly reduced the risk of ischemic cardiovascular events in patients with recent myocardial infarction [101]. Due to vascular occlusion at the infarct site during acute infarction, blood flow cannot pass through, resulting in ischemia and hypoxia at the infarct site, anaerobic enzymolysis of cardiomyocytes and infarcted vascular sites, and the release of a large number of inflammatory factors from the local myocardium, making the infarct site a slightly acidic environment [102]. Wang et al. [103] took advantage of the significant differences in the microenvironment between the infarct site and normal myocardium to synthesize a pH-responsive sustained-release system (ColCaNPs) based on calcium carbonate nanoparticles that could target the delivery of COL to the site of the lesion, thereby improving its efficacy in reducing myocardial infarction and adverse drug reactions. The myocardial infarct size was reduced by approximately 45% in rats treated with ColCaNPs compared to rats treated with no drug intervention after ligation of the anterior descending branch of the left coronary artery MI. This may be due to the ability of COL calcium carbonate nanoparticles to ameliorate myocardial fibrosis by modulating the TLR4/NFKB/NLRP3 signaling pathway and inhibiting cardiomyocyte scorching and inflammatory responses. This delivery system allows for the targeted delivery of colchicine to the lesion and reduces the side effects of the drug.

### 4.2. Intra-Articular Injection

Targeted delivery of high concentrations of therapeutic drugs directly into the joint space using joint cavity injection delivery is a common method used clinically for the treatment of arthritis, which allows the administration of a low dose and minimizes systemic side effects [104,105,106]. Given the therapeutic efficacy of COL in arthritic diseases, researchers have considered COL delivery via intra-articular injections. However, most drug formulations used for intra-articular injection are solution-based injections, which are prone to rapid penetration into the body circulation after administration and may rapidly escape from the joint cavity, resulting in low drug concentration at the target site; this has become a challenge that must be overcome [107]. Thus, new dosage forms, represented by injectable sustained- or controlled-release formulations, are gradually becoming a research hotspot for intra-articular delivery of COL.

Given that the deposition of intra-articular MSU crystals stimulates the production of inflammatory cytokines by macrophages and that CD44, a transmembrane glycoprotein highly expressed on macrophages, binds specifically to the P6 peptide (20 amino acid residue peptides), Zoghebi et al. [108] sought to couple COL to the P6 peptide. The P6 peptide was able to partially bind CD44 and thus reduce the phagocytosis of CD44-mediated MSU, while COL was able to reduce the production of ROS and downstream inflammatory chemokines. Therefore, the coupling reduced both the off-target cytotoxicity of COL and enhanced its anti-inflammatory effect, opening a new avenue for the action of COL-peptide complexes in anti-inflammatory therapy.

Mohamed H et al. [109] developed a COL nanoemulsion system using an aqueous titration technique to solve the problems of narrow COL therapeutic index and low drug concentration at the target site, and the optimized COL nanoemulsion system was screened by establishing a pseudo-ternary phase diagram of oil (isopropyl myristate), surfactant (Span 60), and co-surfactant (ethanol). The 24 h in vitro release rate was 75 ± 0.45%, which was significantly lower than that of the aqueous COL solution (90 ± 0.45%), probably due to the higher encapsulation rate of the COL nanoemulsion system and the fact that drug diffusion had to overcome two barriers: the surfactant interfacial layer and the oil nucleus. In addition, researchers radiolabeled COL with 99 mtc and combined the obtained complexes with an optimized nanoemulsion system for experimental biodistribution studies, which showed that the developed COL nanoemulsion system successfully prolonged the retention time of the drug in the joint cavity and improved its anti-gout activity. Although intra-articular injection is a promising strategy for COL delivery, the discomfort caused by local swelling and pain remains a major obstacle for clinical application, as it leads to reduced patient adherence to treatment. In addition, the stability of the carrier and possible systemic toxic reactions caused by drug leakage and abrupt release are issues of concern for intra-articular COL injections.

## 5. Conclusions

COL has been proposed as a powerful anti-inflammatory agent and is being recently used in many conditions, including colon cancer, atherosclerosis, and pericarditis, in addition to its current medicinal applications (gout, familial Mediterranean fever, leukodystrophy, chondrocalcinosis, and other crystalline arthritis). However, COL is a double-edged sword; although it has good efficacy, its clinical application is limited by frequent toxic reactions and a narrow therapeutic window. To alleviate toxicity and improve efficacy, several novel delivery strategies for COL have been explored. Based on numerous examples in the literature, this review presents a classification of these according to different routes of administration, including oral, injection, and transdermal delivery. While the development of injectable routes with targeted delivery systems and sustained-release oral formulations has led to some breakthroughs in achieving COL toxicity reduction and efficacy enhancement, transdermal delivery has been more extensively investigated for COL delivery because of its advantages of avoiding gastrointestinal digestive reactions after oral administration and pain caused by injectable administration, which is the primary focus of this review. In addition, the permeation-promoting technologies utilized for transdermal drug delivery and their advantages and limitations are briefly discussed.

It is undeniable that nanocarriers have become an important pharmacological means to promote the transdermal absorption of COL. However, it is worth noting that shortcomings such as low bioavailability, toxicity, clearance from the bloodstream, or stimulation of innate immune responses have limited the clinical application of lipid nanocarriers. Polymeric nanocarriers have only been approved by the FDA for a few drugs due to the increased risk of particle aggregation and toxicity [110]. The design and synthesis of some nanomedicines in inorganic nanocarriers are more complicated, the synthesis cost is high, the safety evaluation is inaccurate, the efficacy evaluation system is not better, etc., which has also prevented their clinical translation [111]. Moreover, most nanocarriers have difficulty carrying drugs directly through the stratum corneum, and their permeation facilitation is limited. The combined application of nanocarriers with chemical permeation promoters or physical permeation promotion techniques may achieve better transdermal delivery. However, the effects of the addition of chemical permeation enhancers on the structure and quality of nanocarriers, the invasiveness of microneedles, and the cost of their coupling methods with nanocarriers still need to be addressed by more effective means. We should note that most of the nanocarrier technologies have only laboratory studies and theoretical guidance, and the same is true for colchicine. Future delivery routes and technologies should be more focused on the clinical aspects of the application, and there is still a lot of room for development from theory to practice.

## Figures and Tables

**Figure 1 pharmaceutics-16-00222-f001:**
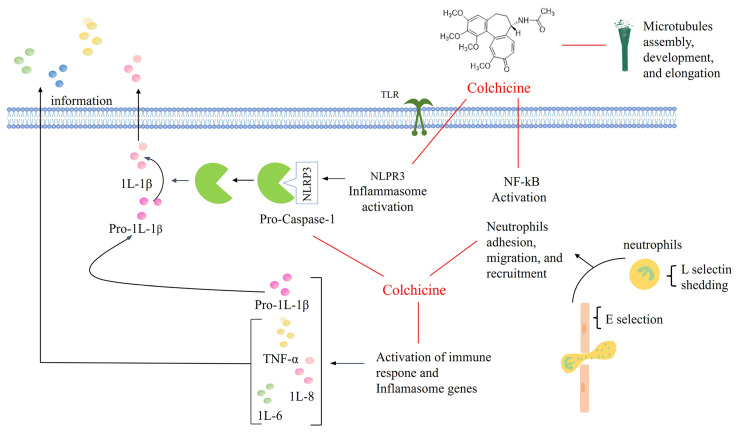
Schematic illustration of the regulatory mechanisms of COL in response to multiple inflammatory responses.

**Figure 2 pharmaceutics-16-00222-f002:**
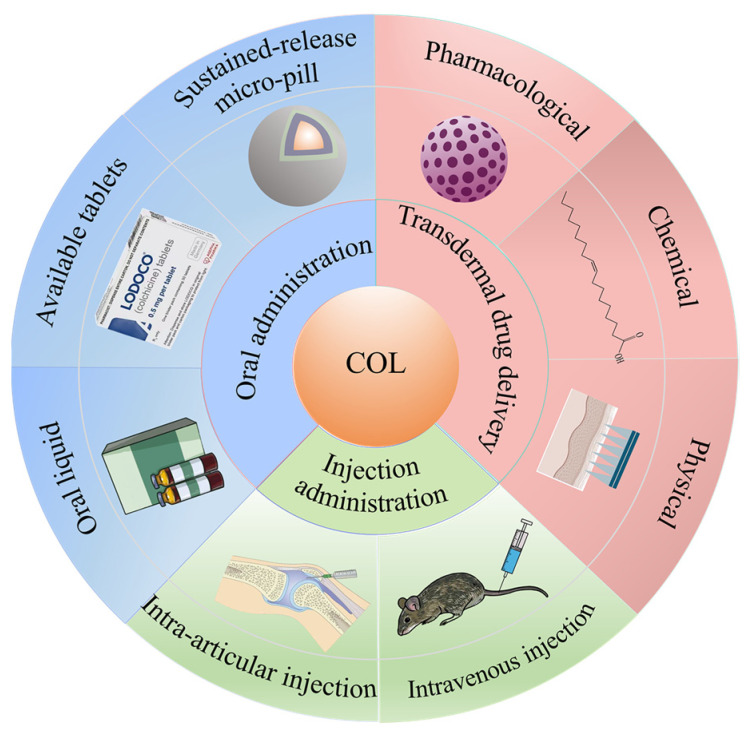
Schematic representation of COL delivery modes using transdermal, injectable, and oral routes.

**Figure 3 pharmaceutics-16-00222-f003:**
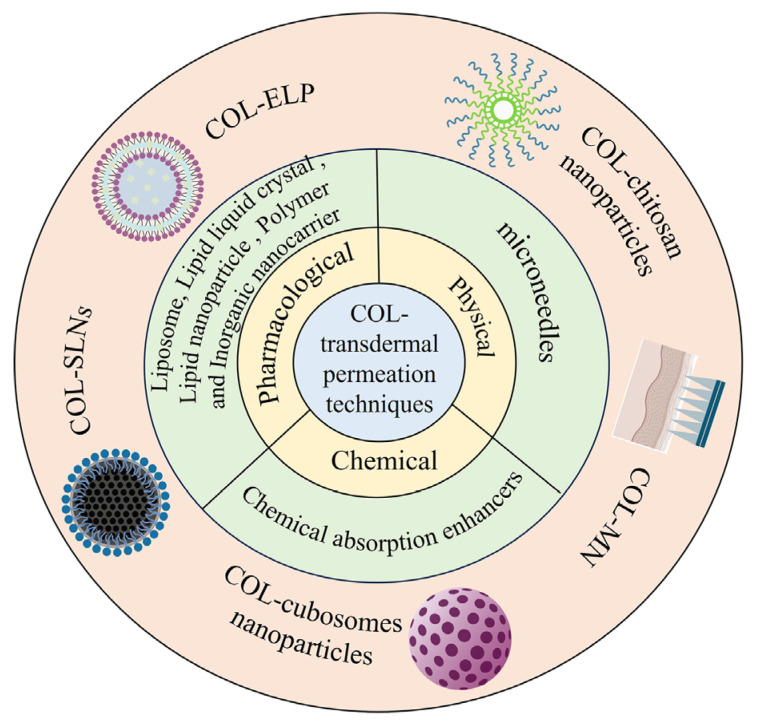
Scheme of COL’s transdermal penetration technology.

**Figure 4 pharmaceutics-16-00222-f004:**
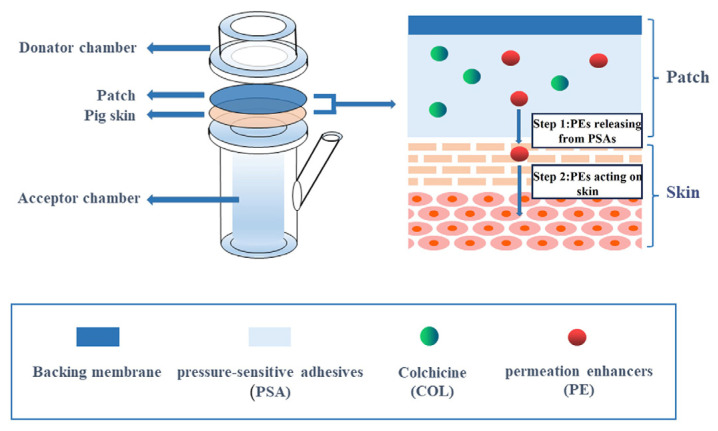
In Vitro transdermal test device and schematic patch release mechanism. (Reprinted with permission from Ref. [32]).

**Figure 5 pharmaceutics-16-00222-f005:**
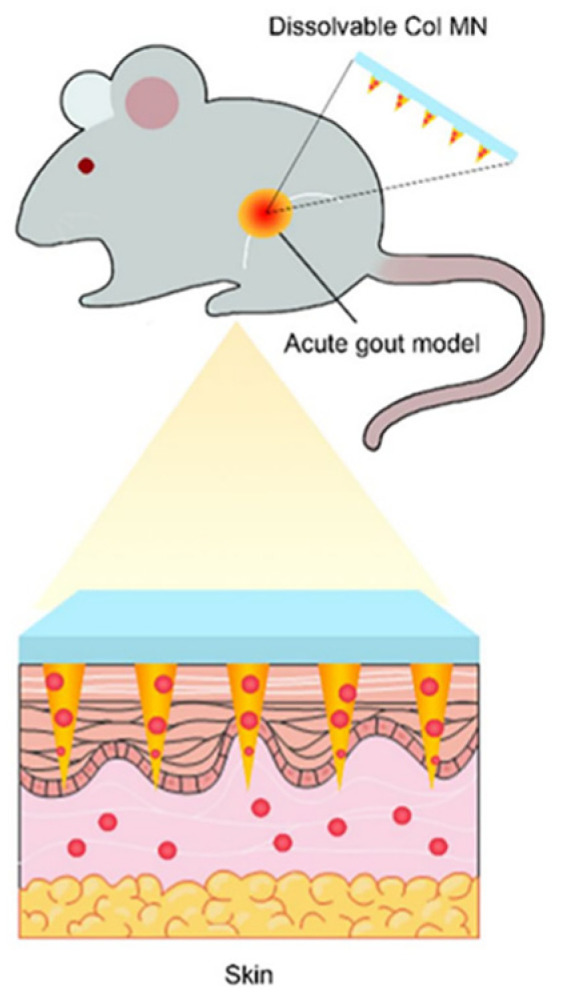
Dissolvable COL-MN introduces COL from the stratum corneum, slowing down gout symptoms in rats. (Reprinted with permission from Ref. [33]. Copyright © 2022 Taylor & Francis group).

**Figure 6 pharmaceutics-16-00222-f006:**
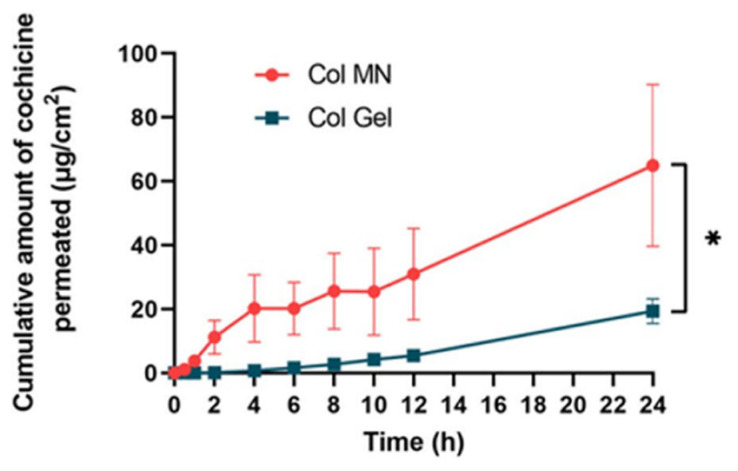
COL-MN showed a 3.36-fold increase over COL gels in drug penetration rate. Both were loaded with equal amounts of active ingredients. * indicates statistical significance at *p* < 0.05, n = 3; (Reprinted with permission from Ref. [33]. Copyright © 2022 Taylor & Francis group).

**Figure 7 pharmaceutics-16-00222-f007:**
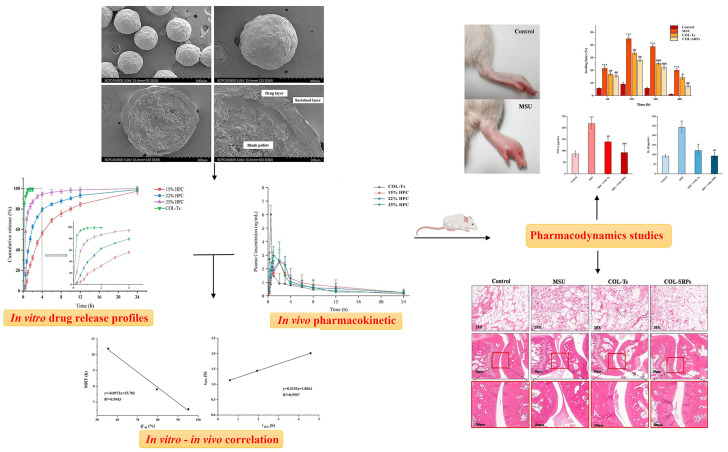
Three COL-SRPs with different in vitro release rates were prepared and characterized. After oral administration, slow-release micropills with 15% HPC content showed better in vivo slow-release properties and better anti-inflammatory effects. The red frame images represent inflammatory cell infiltration into the joint cavity (** *p* < 0.01, *** *p* < 0.001 vs Control. # *p* < 0.05, ## *p* < 0.01, ### *p* < 0.001 vs MSU) (Reprinted with permission from Ref. [95]. Copyright © 2021 Elsevier).

**Figure 8 pharmaceutics-16-00222-f008:**
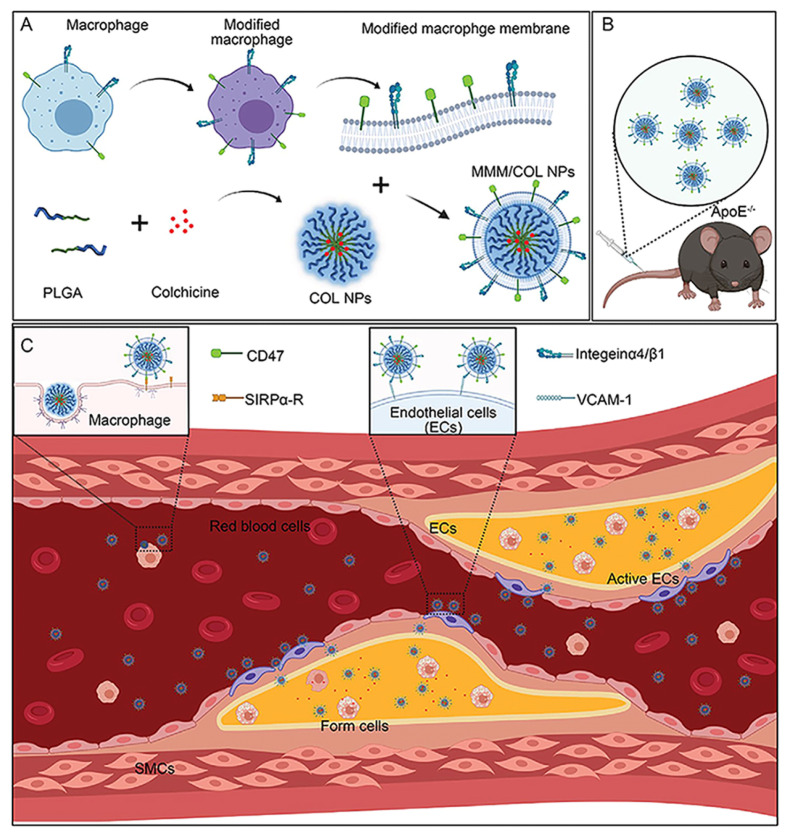
Schematic Mechanism for Preparation of MMM/COL NPs and MMM/COL-NP Delivery of Colchicine for the Treatment of Atherosclerosis. (**A**) Schematic illustration of the preparation of MMM/COL NPs. (**B**) Schematic diagram of intravenous injection of MMM/COL NP to the vulnerable atherosclerotic plaque mice. (**C**) Schematic diagram of immune evasion and atherosclerotic plaque targeting performance of MMM/COL NP in the blood vessels (Reprinted with permission from Ref. [97]. Copyright © 2021 Elsevier).

**Table 1 pharmaceutics-16-00222-t001:** Common transdermal permeation techniques for COL.

Transdermal Osmotic Technology	Types	Benefits	Carriers	Reference
Pharmacological permeation promotion technology	Liposome technology	Favorable skin affinity and enhancing stratum corneum hydration	COL-ELP	[23]
			COL-βCD-ELP	[23]
			COL-alcohol plasmids	[24]
			COL-alcohol transporters	[25]
	Niosomes technology	Easily accessible and high stability	COL-vesicles	[26]
	Lipid liquid crystal technology	Multi-drug loading with high capacity	COL-cubosomes nanoparticles	[27]
	Lipid nanoparticle technology	Strong lipophilicity and small particle size	COL-SLNs	[28]
	Polymer nanocarrier technology	Solubilizes and improving release rate	COL-chitosan nanoparticles	[29]
			COL-PPCF-NPs	[30]
	Inorganic nanocarrier technology	High drug loading capacity and controllability	COL-MSNs	[31]
Chemical permeation promotion technology	Chemical absorption enhancers	Simple structure and high compliance for PSA	COL-DIA patch	[32]
Physical permeation promotion technology	microneedles	Expanding the range of drugs and reducing side effects	COL-MN	[33]
			UAO-LPO/COL-MN	[34]
			COL-HMNs	[35]

## Data Availability

Not applicable.

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
