# Peer review of "Delivery Strategies for Colchicine as a Critical Dose Drug: Reducing Toxicity and Enhancing Efficacy"

_pharmaceutics, 2024, doi:10.3390/pharmaceutics16020222_

Round 1
Reviewer 1 Report
Comments and Suggestions for Authors
The manuscript entitled: “Delivery Strategies for Colchicine as a Critical Dose Drug: Reducing Toxicity and Enhancing Efficacy” fits well into the aims and scope of Pharmaceutics.
General:
This review article discusses advancements in research related to various delivery methods aimed at reducing toxicity and improving effectiveness. It highlights utilizing the transdermal drug delivery route to avoid metabolic impact and alleviate the discomfort often associated with oral and injection methods. Also, the authors discuss several techniques that can be used in colchicine delivery. Furthermore, this article briefly explores the permeation-promoting technologies employed in transdermal drug delivery, along with their respective advantages and limitations, including nanocarriers, which have emerged as a significant pharmacological approach to enhance the transdermal absorption of COL. The conclusion covered the most important points mentioned in the article. Recent references are used with few numbers of self-publication. So, the article handles an important point and is well-written however, some corrections are required in the comments for authors:
Comments:
1- Some abbreviations are missed so please add a list of abbreviations.
2- Line 42, “NSAIDs” please mention the meaning of each abbreviation first then you can use it.
3- Figures 1, 2, and 3 without reference. Are these figures your own or cited from others? If you cited, please add the reference for each figure.
4- Figure 7 is not clear and should be improved
5- The article lacks details about natural nanocarriers.
6- Minor English revision.
Comments on the Quality of English LanguageMinor revision
Reviewer 2 Report
Comments and Suggestions for Authors
I have received a review Manuscript (pharmaceutics-281914) entitled: ‘Delivery strategies for colchicine as a critical dose drug reducing toxicity and enhancing efficacy’ submitted to Pharmaceutics.
In the elaborated review article the authors gathered and summarized recent advances in research progress of different delivery strategies for colchicine in terms of toxicity reduction and efficacy enhancement. I consider this topic as highly valuable as there are no other similar reviews in this topic. Generally, this Manuscript is well-written and summarizes the-state-of-the-art in a very interesting way. All paragraphs are elaborated with care and do not bring any confusion to reader. I noticed that authors have included a short introduction at the beginning of each chapter, providing all the basic information that can turn out to be very useful for readers less familiar with this topic. In effect, such revision will in future contribute to development of further research focused on effective colchicine delivery formulations. It is an excellent candidate for publication in Pharmaceutics, however, I advise to:
1. improve the clarity of the Figures 2 and 3 – they are too small (particularly font size), which makes it difficult to study
2. add a column in table 1 summarizing the most important information (benefits, and maybe even an increase in efficiency compared to a pure substance) about the technology used
3. lines 215-242: the abbreviations (TET, TENa and TEL) should be explained at the beginning of this paragraph
4. Conclusions should be enriched with 3-4 additional sentences demonstrating the most crucial perspectives for future development of colchicine delivery formulations
I also recommend to add one reference and describe in a few sentences the significance of this recently published review article in the introduction section: Lisa K.Stamp et al., Colchicine: the good, the bad, the ugly and how to minimize the risks, Rheumatology, 2023, https://doi.org/10.1093/rheumatology/kead625
Reviewer 3 Report
Comments and Suggestions for Authors
In Abstract
Lines 13-14 The sentence its clinical application is limited by its gastrointestinal adverse effects and serious gastrointestinal adverse effects, and only oral formulations are currently marketed worldwide contains a repetition. In any case, the toxicity of COL is underestimated in all the MN, despite the large literature about this argument. The reader should more informed about this aspect and the relative reports about the risk/benefit argument. As an example, the sentence in the Conclusion However, COL is a double-edged sword; although it has good efficacy, its clinical application is limited by frequent toxic reactions and narrow therapeutic window, reporting also the important studies about this argument.
In Introduction
About the sentence Colchicine (COL), a tricyclic alkaloid extracted from the lily plant Colchicum. The lily plant is Colchicum autumnalis L. , Colchicum is the genus and in any case the botanical name of the species must be in italic. However, COL is also obtained from other species, like in the European Pharmacopoeia where is reported Gloriosa superba L:
Line 42-44 the sentence Unlike NSAIDs and steroids that act on the arachidonic acid pathway, it binds to the cytoskeletal protein microtubule protein dimer, inhibiting microtubule aggregation at low concentrations and promoting microtubule depolymerization at high concentrations [8] needs an explanation to the reader of the meaning of NSAIDs and the consequence related to the inhibition of microtubules, as well the argument related to problems about the utilization of COL in cancer therapy.
Line 46-49 About the sentence In vitro studies have shown that at nanomolar concentrations, COL can reduce the expression of E-selectin on the surface of endothelial cells and inhibit neutrophil adhesion to inflammatory tissues, whereas at micromolar concentrations, COL inhibits the expression of the adhesion molecule L- selectin on neutrophils, further preventing their recruitment [9] it should be better explained the meaning of nanomolar, although most of the MN is dedicated to this argument.
Line 60 About the sentence Although COL is a natural product with multiple biological activities and COL formulations have been marketed internationally as early as the 1960s it should be supported by references and considered that COL is considered a potentially toxic compound,
The last part of the Introduction the sentences In addition, the combination of targeted drug delivery systems to effectively deliver COL to the disease site by injection and the preparation of extended-release formulations for oral administration to achieve stable blood concentrations and thus reduce side effects are also effective means of reducing drug toxicity and enhancing efficacy [16]. Therefore, this review classifies and summarizes the main delivery strategies of COL in recent years based on different delivery routes that may provide new ideas and discussions for the development of formulations to reduce COL toxicity and enhance its efficacy contains a repetition of the same concepts.
Inflammatoy states are usually associated with oxidative stress. Therefore, a comment about the antioxidant activity of COL should be better reported in page 16, including some references.
In the Conclusion the last sentence, which should be the significance of the MN reports: Although most technologies still have some limitations, it is believed that with the continual refinement of nanocarrier delivery theory and technology, safer and more efficient formulations will be developed. However, it could be interesting to evidence the most relevant limitations, being the “some” very generic, and the “it is believed” should be revised in consideration of the reader. Why it is believed? Who believe? The authors? Why the refinement should be the main aspect on the formulation sector? Unfortunately, the history of pharmaceutical drugs tell us several negative episodes about safety and efficiency.
Comments about a comparison between the real possibilities of each technique and products to realize marketed drugs should be reported. It should be also considered that most of the current research in nanotechnology, including in the sector reported in the MN, is related to lab experiments and there is still a long way to reach ordinary people.
The general final comment is that a review should overcome the simple list of the researches and experiments already performed and present in the literature, but it should be offer to the reader comments and comparisons, as well competent information about the future.
